# Abnormal Iron and Lipid Metabolism Mediated Ferroptosis in Kidney Diseases and Its Therapeutic Potential

**DOI:** 10.3390/metabo12010058

**Published:** 2022-01-10

**Authors:** Xiaoqin Zhang, Xiaogang Li

**Affiliations:** 1Department of Internal Medicine, Mayo Clinic, Rochester, MN 55905, USA; 2Department of Nephrology, Tongji Hospital, School of Medicine, Tongji University, Shanghai 200065, China; 3Department of Biochemistry and Molecular Biology, Mayo Clinic, Rochester, MN 55905, USA

**Keywords:** abnormal metabolism, ferroptosis, kidney disease, therapeutic

## Abstract

Ferroptosis is a newly identified form of regulated cell death driven by iron-dependent phospholipid peroxidation and oxidative stress. Ferroptosis has distinct biological and morphology characteristics, such as shrunken mitochondria when compared to other known regulated cell deaths. The regulation of ferroptosis includes different molecular mechanisms and multiple cellular metabolic pathways, including glutathione/glutathione peroxidase 4(GPX4) signaling pathways, which are involved in the amino acid metabolism and the activation of GPX4; iron metabolic signaling pathways, which are involved in the regulation of iron import/export and the storage/release of intracellular iron through iron-regulatory proteins (IRPs), and lipid metabolic signaling pathways, which are involved in the metabolism of unsaturated fatty acids in cell membranes. Ferroptosis plays an essential role in the pathology of various kidneys diseases, including acute kidney injury (AKI), chronic kidney disease (CKD), autosomal dominant polycystic kidney disease (ADPKD), and renal cell carcinoma (RCC). Targeting ferroptosis with its inducers/initiators and inhibitors can modulate the progression of kidney diseases in animal models. In this review, we discuss the characteristics of ferroptosis and the ferroptosis-based mechanisms, highlighting the potential role of the main ferroptosis-associated metabolic pathways in the treatment and prevention of various kidney diseases.

## 1. Introduction

The kidney is a complex organ consisting of a million filters and miles of vasculature that are responsible for maintaining multiple aspects of homeostasis, including managing fluid levels, electrolyte balance, and other factors that keep the internal environment of the body consistent and comfortable in the human body. To successfully perform these crucial functions, numerous cell types are arranged in the complex three-dimensional structure of the nephron, responding to a variety of hormonal, neuronal, inflammatory, and intra- and intercellular signals. The kidney is susceptible to a wide range of abnormalities from structural defects to functional dysfunction due to various conditions, including genetic variation, environmental, sociodemographic, and clinical factors. Kidney diseases, ranging from acute kidney disease, such as acute kidney injury (AKI), to chronic kidney disease (CKD), such as diabetic kidney disease (DKD) and autosomal dominant polycystic kidney disease (ADPKD), are a global public health problem, affecting over 750 million persons worldwide [1]. Thus, understanding the molecular mechanisms underlying kidney diseases is important and crucial for drug development and the generation of novel therapeutic strategies. In this review, we discuss the roles and mechanisms of one of the novel regulated cell deaths, ferroptosis, in kidney diseases.

Regulated cell death is an indispensable biological process, including apoptosis and necrosis, autophagy, and ferroptosis, which is essential for normal development and the maintenance of homeostasis as well as preventing the onset of diseases. Ferroptosis as a novel regulated cell death is defined by the accumulation of lethal lipid species derived from lipid peroxidation and subsequent membrane damage [2]. Ferroptotic cell death is also accompanied by a series of variations in cell morphology that contributes to discriminating it from other forms of cell death. At the cellular and subcellular levels, ferroptosis does not have the morphological characteristics of typical apoptosis, such as cell shrinkage, chromatin condensation, and formation of apoptotic bodies, and disintegration of the cytoskeleton, nor double-membrane enclosed vesicles (autophagic vacuoles) from autophagic cells. In addition, morphological features, such as swelling of the cytoplasm and organelles and rupture of the cell membrane from necrotic cells are not observed in ferroptotic cells. The nuclei in ferroptotic cells conserve its structural integrity, the cell membrane remains intact, while the mitochondria manifest shrinkage with increasing membrane density and the reduction or disappearance of mitochondrial cristae [2]. The morphological changes in mitochondria in ferroptotic cells are a different process from other modes of cell death [3,4]. We summarized the main morphological and biochemical features of the different types of cell death in Table 1. Growing evidence support that ferroptosis plays a critical role in different kidney diseases, including AKI, CKD, and RCC (Renal cell carcinoma). Ferroptosis can be triggered by the inhibition of system Xc- (e.g., Erastin, Salazosulfapyridine, Sorafenib) and glutathione peroxidase 4 (GPX4) (e.g., RSL3/5, FINs, FINO2, DPIs), starvation of cysteine, and peroxidation of arachidonoyl (AA), whereas antioxidants scavenge lipophilic radical (e.g., Ferrostatin-1, Liproxststatin-1, Vitamin E), iron chelators (e.g., Deferoxamine, Deferoxamine mesylate), and deuterated polyunsaturated fatty acid phospholipids (e.g., Rosiglitazone, Pioglitazone, Troglitazone) can prevent ferroptosis [5,6,7]. In this review, we focus on an in-depth discussion of the mechanisms and regulation of ferroptosis, by highlighting the advances made through different approaches to the development of targeted therapies linked to potential effects on improving outcomes for patients with kidney diseases.

## 2. The General Mechanisms of the Regulation of Ferroptosis

Ferroptosis can be regulated by a set of factors and signaling pathways associated with a variety of metabolic changes, including abnormal amino acid metabolism, the accumulation of iron, and subsequent lipid peroxidation. In this section, we discuss the key regulators involved in the ferroptotic process and summarize the basic mechanisms and metabolic networks of ferroptosis (Figure 1).

### 2.1. The Regulation of Ferroptosis by the GSH/GPX4 Pathway

System Xc- is a widely occurring amino acid antiporter in the phospholipid bilayer, and is a heterodimer composed of two subunits, solute carrier family 7 member 11 (SLC7A11) and solute carrier family 3 member 2 (SLC3A2) [8]. System Xc- controls the cellular entry and exit of cystine (Cys) and glutamate (Glu), respectively, at a ratio of 1:1 [9]. Cys is reduced to cysteine in cells, which influences the synthesis of glutathione (GSH). GSH activates GPX4 and influences intracellular redox homeostasis. Inhibition of the activity of the system Xc- affects GSH synthesis by decreasing the absorption of Cys, resulting in a decrease in GPX4 activity, a decline in cell antioxidant capacities, lipid reactive oxygen species (ROS) accumulation, and ultimately the occurrence of oxidative damage and ferroptosis (Figure 1, left upper). P53 is an important tumor suppressor gene, which can downregulate the expression of SLC7A11 to negatively regulate system Xc- mediated uptake of cystine (Figure 1, left upper). The reduction in Cys absorption further affects the activity of GPX4, resulting in a reduction in cell antioxidant capacity, accumulation of lipid ROS, and leading to ferroptosis [10,11].

GPX4 is one of eight well-known GSH peroxidases in mammals [12] and one of the most important and studied enzymes in the ferroptotic process. It reduces complex hydroperoxides, including phospholipid hydroperoxides and cholesterol hydroperoxides to their corresponding counterparts, thereby interrupting the lipid peroxidation chain reaction [13]. GPX4 is a selenoprotein and contains a selenocysteine (Sec) at its active site and its catalytic function requires GSH for regeneration of its active state. When GSH is depleted, GPX4 activity is reduced or inactivated (Figure 1, left upper). GPX4 can also convert GSH into glutathione disulfide (GSSG), resulting in the reduction in esterified oxidized fatty acids and cholesterol hydroperoxides, and the decrease in lipid hydroperoxide (L-OOH) to a nontoxic lipid hydroxy derivative (L-OH), which resist oxidative damage [14]. Both erastin and RSL3 induce ferroptotic cell death by inactivating GPX4 [6]. A recent study showed that erastin not only inactivated GPX4 but could also repress GPX4 expression by upregulating the expression of ATF3, in which ATF3 could repress the expression of GPX4 [15].

### 2.2. The Regulation of Ferroptosis by Iron Metabolic Pathways

Iron metabolism is regulated by a perfectly adjusted balance between plasma proteins and is thought to be one of the central mediators of ferroptosis [16]. These plasma proteins are associated with the transport, absorption, and recycling of iron. Circulating iron exists in the form of ferric iron (Fe^3+^) bound to transferrin. Fe^3+^ is introduced into the cell via the membrane protein transferrin receptor 1 (TFR1) and then localized to the endosome [17] (Figure 1, right upper). Iron reductase reduces Fe^3+^ to ferrous iron (Fe^2+^) in the endosomes, which can be released from endosomes into unstable iron pools in the cytoplasm by divalent metal transporter 1 (DMT1) or Zinc–Iron regulatory protein family 8/14 (ZIP8/14). This internal iron recycling strictly controls iron homeostasis in cells. It has been reported that overexpressed heat shock protein beta-1 (HSPB1) can reduce intracellular iron concentrations by inhibiting TRF1 expression, resulting in the inhibition of ferroptosis [18]. In addition, excessive iron can be stored in ferritin/hemosiderin or remain labile. Ferritin is an iron storage protein complex that consists of ferritin light chain (FTL) and ferritin heavy chain 1 (FTH1) [4]. FTH has iron oxidase activity, which catalyzes the conversion of the Fe^2+^ to the Fe^3+^, allowing iron to be safely incorporated into the ferritin shell, thereby reducing the free iron levels [19]. The Fe^3+^/Fe^2+^ redox potential is involved in many protein complexes, especially those that involve oxygen reduction for adenosine triphosphate (ATP) synthesis and the reduction in DNA precursors. Studies have been demonstrated that the nuclear receptor coactivator 4 (NCOA4)-mediated ferritinophagy played a vital role in the release of iron from ferritin. NCOA4 binds to ferritin and delivers it to lysosomes for degradation (Figure 1, right upper). Further, the degradation releases iron and increases the abundance of iron in the cell. Therefore, several studies indicated that NCOA4-mediated ferritinophagy promoted ferroptosis as a result of an increase in the availability of intracellular iron. Hydrogen peroxide (H_2_O_2_) can react with ferrous ions and produce hydroxyl radicals with strong oxidizing properties. This reaction is called the Fenton reaction [20] (Figure 1, right upper). The Fenton reaction has high reactivity with biological molecules, such as proteins and DNA, which generates lipid (hydro) peroxidation in ferroptosis [21,22].

### 2.3. The Regulation of Ferroptosis by the Lipid Metabolic Pathways

Lipid peroxidation, which is the hallmark of ferroptosis, is the archetype free radical chain reaction that formally results in the insertion of oxygen into the C-H bond in the oxidizable free polyunsaturated fatty acids (PUFAs), which involved in ferroptosis in all pathways. Lipid peroxidation can be divided into three phases: initiation, propagation, and termination. Any radical that can abstract an H-atom from an oxidizable substrate, such as PUFAs, can initiate the lipid peroxidation process. The abundance and location of intracellular oxidizable substrates of lipid peroxidation determine the extent of lipid peroxidation and ferroptosis. Free polyunsaturated fatty acids are the substrate of the synthetic lipid signal transduction medium, which can be esterified into membrane phospholipids in lipid metabolism. Lipidomic analyses indicate that phosphatidylethanolamines (PEs) containing arachidonic acid (AA) or adrenic acid (AdA) are the key membrane phospholipids, which oxidized phospholipid hydroperoxides (PE-AA/AdA-OOH) by non-enzymatic processes, such as free radical lipid peroxidation or Fenton chemistry, drive ferroptosis. Acyl-CoA synthetase is a long-chain family member 4 (ACSL4) and lysophosphatidylcholine acyltransferase 3 (LPCAT3) are two enzymes that participate in the biosynthesis and remodeling of PE, activate PUFAs, and affect the transmembrane characteristics of PUFAs (Figure 1, left bottom). AA/AdA is acylated into membrane phospholipids by LPCAT3 and ACSL4 (PE-AA/AdA) [23,24]. Therefore, blocking the expression of ACSL4 and LPCAT3 results in the suppression of AA or AdA esterification into PE reducing the accumulation of lipid peroxide substrates in cells, thus inhibiting ferroptosis [24]. Conversely, cells supplemented with arachidonic acid or other polyunsaturated fatty acids are sensitized to ferroptosis [25].

Lipoxygenases (LOXs) is a non-heme iron-containing protein, which can mediate ferroptotic peroxidation. Free polyunsaturated fatty acids, rather than polyunsaturated fatty acid-containing phospholipids, are the preferred substrates of LOXs. It was found that genetic depletion of LOXs protects against erastin-induced ferroptosis [25], suggesting LOXs’ ferroptosis contribution.

### 2.4. The Regulation of Ferroptosis by the Mitochondrial Metabolic Pathways

Mitochondria are double membrane-bound organelles that conduct oxidative phosphorylation and generate most of the energy and play a central role in cell death regulation. However, studies suggested that the role of the mitochondrion in ferroptosis is complex and somewhat controversial [26]. On the one hand, the mitochondria play a pro-ferroptosis function. The mitochondrion is the major organelle to produce cellular ROS. Leakage of electrons at mitochondria complex I (NADH: ubiquinone oxidoreductase) and complex III (coenzyme Q: cytochrome c–oxidoreductase) from electron transport chains (ETC) leads to superoxide (O_2_•−) production, which is subsequently dismutated to H_2_O_2_ through superoxide dismutase (SOD). H_2_O_2_ can react with Fe^2+^ to generate hydroxyl radicals (•OH), which abstract the bisallylic hydrogen from PUFAs to generate PUFA radicals (PUFA•). PUFA• further react rapidly with oxygen to form PUFA peroxyl radicals (PUFA-OO•) and form PUFA hydroperoxides (PUFA-OOH) ultimately. Therefore, the production of mitochondrial ROS actively drives ferroptosis by promoting lipid peroxidation. In addition to ROS production, other important metabolism processes, such as ATP synthase and the tricarboxylic acid (TCA) cycle, suggested that mitochondrion plays an important role in inducing ferroptosis. For example, ETC activity has been shown to promote cysteine-deprivation-induced ferroptosis, and treatment with various ETC complex inhibitors or mitochondrial uncoupling agents has been shown to block ferroptosis [27]. In addition, this mechanism, possibly through energy depletion, subsequently inactivated the energy sensor AMP-activated protein kinase (AMPK) signaling pathway [28]. Voltage-dependent anion channels (VDACs) are the transmembrane channels that transport ions and metabolites. It is reported that erastin acts on VDACs, leads to mitochondrial dysfunction, and results in the appearance of oxidative species, and eventually leads to iron-mediated cell death [29] (Figure 1, right bottom). On the other hand, despite accumulated evidence supporting an important role of mitochondria in driving ferroptosis, it is worth noting that the study from GPX4 inactivation-induced ferroptosis is not accompanied by obvious mitochondrial lipid peroxidation [30]. Consistent with this, treatment with the mitochondrion-specific ROS scavenger MitoTEMPO had a limited protective effect on cotylenin A and phenethyl isothiocyanate-induced ferroptotic cell death in pancreatic cancer cells [31].

### 2.5. The Regulation of Ferroptosis by Other Signaling Pathways

Ferroptosis can be also regulated by other pathways, such as the mevalonate pathway and sulfur transfer pathways. First, the toxic small-molecule ferroptosis-inducing 56 (FIN56) is required for mevalonate pathway-mediated ferroptosis. FIN56 can activate its own target protein squalene synthase besides inducing ferroptosis by decreasing the abundance of GPX4 [32]. Second, methionine can be converted into cystine through the sulfur transfer pathway under oxidative stress, and then GSH can be synthesized to further exert its antioxidant effects. Third, the p53 regulatory axis performs dual regulatory functions in ferroptosis. On the one hand, p53 can enhance ferroptosis by inhibiting the expression of SLC7A11 or enhancing that of spermidine/spermine N1-acetyltransferase 1 (SAT1) and glutaminase 2 (GLS2) [33] (Figure 1, left bottom). On the other hand, p53 suppresses ferroptosis by reducing the activity of dipeptidyl peptidase-4 (DPP4) and increasing the expression of cyclin-dependent kinase inhibitor 1A (CDKN1A) [34] (Figure 1, left bottom). Fourth, the ferroptosis suppressor protein 1 (FSP1)–coenzyme Q10-nicotinamide adenine dinucleotide phosphate (FSP1-CoQ10-NADPH) axis exists as a parallel system, which co-operates with GPX4 and glutathione to suppress phospholipid peroxidation and ferroptosis [35] (Figure 1, right bottom). Last, the p62-Keap1-Nrf2 [36,37], and glutamine metabolic pathways can effectively regulate the formation of intracellular irons and ROS acts as a regulatory role in ferroptosis (Figure 1, right upper).

## 3. Ferroptosis and Kidney Diseases

### 3.1. The Role of Ferroptosis in AKI

AKI, also known as acute renal failure, is defined as an abrupt (within hours) decrease in kidney function, ranging from minor loss of kidney function to complete kidney failure [2]. AKI is common in patients who are critically ill and already in the hospital especially elderly patients in intensive care units. AKI diagnostic is based on the acute reduction in glomerular filtration rate (GFR) with time intervals [2]. AKI can be caused by a reduction in blood flow, direct injury to the kidneys, and urinary tract block [3]. Signs and symptoms of AKI differ depending on the cause, such as not enough urine, swelling in legs, ankles, or around the eyes, and fatigue or tiredness. Generally, vasoconstriction, oxidative stress, apoptosis, inflammation, and hypoxia are the major pathogenic mechanisms of AKI [38]. In the past, apoptosis was considered as the main regulated cell death in various models of ischemic injury. However, later discoveries suggested that ferroptosis might be a major driver of AKI induced by either ischemia–reperfusion injury (IRI) or oxalate crystal, which is directly involved in the synchronized necrosis of renal tubules [39]. IRI is the main cause of AKI in patients who have undergone cardiac surgery in the clinic. It has been reported that low levels of intraoperative iron-binding proteins, such as serum ferritin, and higher transferrin saturation may reflect an impaired ability to rapidly process catalytic iron released during extracorporeal circulation and lead to kidney damage [40]. The importance of iron homeostasis and ferroptosis in human IRI suggests a potential therapeutic target in cardiac surgery-associated kidney injury or IRI-induced AKI.

GPX4 is the key regulator of ferroptosis that can be degraded by chaperone-mediated autophagy (CMA). It has been shown that the inhibition of CMA can reduce ferroptosis by stabilizing GPX4 [41]. Legumain is conserved as asparaginyl endopeptidase that is highly expressed in proximal tubular cells and promotes chaperone-mediated autophagy of GPX4 [42]. It has been reported that legumain deficiency could ameliorate tubular cell ferroptosis through stabilizing GPX4 in an AKI animal model induced by IRI or nephrotoxic folic acid. In another, AKI model induced by cisplatin, vitamin D receptor activation protected against cisplatin-induced renal injury by inhibiting ferroptosis partly via trans-regulation of GPX4 [43]. In addition, the incidence and mortality in spontaneous AKI were significantly increased in GPX4 knocked mice [30]. The survival of the GPX4-deficient mice could be extended by approximately 35% by the clearance of lipid peroxides in vivo [30]. These findings suggest that inhibition of GPX4 may play an important role in ferroptosis-associated kidney injury.

Due to the role of iron in mediating the production of reactive oxygen species and enzyme activity in lipid peroxidation, iron metabolism is thought to be one of the central mediators of ferroptosis. The regulation of iron metabolism can suppress inflammation, oxidative stress, and cell damage caused by iron overload and iron disorders. Heme oxygenase-1 (HO-1) is a rate-limiting enzyme in the degradation of heme, which is precursive to hemoglobin and constitutes 95% of functional iron in the human body. Heme can be degraded into bilirubin, CO, and Fe^2+^. Upregulation of HO-1 enhances the degradation of heme and the synthesis of ferritin, altering the intracellular iron distribution [44]. As a dual regulator in iron and ROS homeostasis, HO-1 is suggested to serve a dominant role in ferroptosis [45]. The expression of HO-1 is upregulated in response to oxidant stress in proximal tubular cells (PTCs) in vitro, which exerts cytoprotective and anti-inflammatory effects [46,47]. Knockdown HO-1 are highly sensitive to erastin- and RSL3-induced ferroptosis, whereas treatment with ferrostain-1 (Fer-1) attenuates cellular stress and death in PTCs undergoing ferroptosis in vitro. Moreover, the protective role of HO-1 has also been demonstrated in several animal models of AKI [48,49,50,51]. For example, in the AKI model induced by rhabdomyolysis, induction of HO-1 before injury has resulted in a significant attenuation of structural damage, prevention of kidney failure, and decreasing mortality. In the AKI model induced by cisplatin, HO-1 (^−/−^) mice developed more severe renal failure compared with HO-1 (^+/+^) mice. These findings suggested that HO-1 induction may play an important role in conferring protection on renal cells from oxidative damage and has an anti-ferroptosis effect in renal epithelial cells.

Rhabdomyolysis accounts for 15% of all causes of AKI and the accumulation of myoglobin in the kidney is the key mechanism leading to kidney damage [52]. Myoglobin can be degraded in the kidney, resulting in the release of iron, which participates in the production of oxidizing substances through the catalytic action of the Fenton reaction and the induction of lipid peroxidation in proximal tubular epithelial cells in the AKI animal model induced by rhabdomyolysis [53]. Treatment with iron chelator deferoxamine could alleviate renal injury induced by rhabdomyolysis in vivo and prevent cytotoxicity in vitro [54,55]. These findings suggested that iron-dependent ferroptosis may play an important role in rhabdomyolysis-induced renal injury. Hepcidin, a major regulator of iron homeostasis that prevents iron export from cells by inducing the degradation of the iron export protein, ferroportin [56]. Ferroportin is located at the basolateral membrane of the proximal tubules in the kidneys [57]. The expression of ferroportin is increased in response to export hepatosplenic iron, which results in the alterations of systemic iron homeostasis, including hepatosplenic iron depletion, and increased serum and kidney nonheme iron levels in the renal IRI animal model [58]. Knockout of ferroportin (^−/−^) increased the expression of FTH1, resulting in the chelation of iron and the inhibition of ferroptosis to alleviate ischemic acute kidney injury [57]. Pretreatment with hepcidin prevents renal IRI-induced dysregulation of systemic iron homeostasis and reduces inflammation in AKI animal models [58]. These findings suggest that the hepcidin–ferroportin pathway holds promise as a novel therapeutic target in the treatment of AKI.

Lipid peroxidation is responsible for intense vasoconstriction and oxidative injury, which is an unfavorable factor for AKI [59,60]. Inhibition of lipid peroxidation can alleviate kidney injury induced by cistplatin [61]. It has been reported that treatment with lipophilic antioxidants, diphenyl-p-phenylenediamine (DPPD), and deferoxamine suppresses the accumulation of thiobarbituric acid reactive substances in proximal tubules injured by tert-butyl hydroperoxide (tBHP) [62]. Quercetin, a natural flavonoid, has been shown to block the typical morphologic changes of ferroptotic cells by reducing the levels of malondialdehyde (MDA) and lipid ROS as well as increasing the levels of GSH in proximal tubular epithelial cells [63]. Treatment with quercetin abrogated lipid ROS and protected functional acute renal failure and structural organ damage in AKI mice [63]. In sum, all the studies directly or indirectly support an essential role of ferroptosis in AKI.

### 3.2. The Roles of Ferroptosis in CKD

CKD, also known as chronic kidney failure, is defined as the presence of kidney damage persisting over a long period of time. CKD can cause wastes to build up in the body and is a worldwide public health problem with adverse outcomes, including loss of kidney function, cardiovascular disease (CVD), and premature death. The causes of CKD might vary globally. The two main causes are diabetes and hypertension. Others could include primary glomerulonephritis, inherited diseases, such as polycystic kidney disease, malformations, obstructions caused by problems, such as kidney stones, and repeated urinary infections [64,65].

Iron homeostasis is maintained by multiple mechanisms, such as hepcidin and iron regulatory proteins (IRPs), at the systemic and cellular levels. In normal human kidneys, IRPs in proximal tubules regulate tubular iron handling to avoid iron accumulation. However, in CKD patient kidneys, iron is deposited and results in increased iron uptake and/or inadequate iron export [66]. The accumulation of irons initiates Fenton-mediated oxidative damage and may contribute to renal injury [67,68], suggesting that the CKD kidney iron accumulation is initially inducing ferroptosis and iron plays a detrimental role in the progression of CKD. Therefore, the regulation of iron metabolism proteins is of great significance in restoring kidney iron metabolism and mitigating ferroptosis.

Diabetic kidney disease (DKD) is the most common etiology of chronic kidney disease and accounts for most mortality in patients with diabetes mellitus [69]. Diabetic nephropathy (DN), a severe microvascular complication of diabetes, is characterized by proteinuria and a progressive decline in kidney function, leading to the end-stage of kidney disease [70]. The diabetic kidney is exposed to high glucose, oxidative stress, and advanced glycation end products, which contribute to the progression of nephropathy by inducing glomerular cell activation and inflammatory infiltration [71]. Accumulative studies have indicated that in addition to many forms of programmed cell death, such as autophagy, apoptosis, and necrosis, ferroptosis also plays a pathological role in the development of DN [72]. In diabetic nephropathy of kidney biopsy tissues, the ferroptosis-related molecules SlC7A11 and GPX4 are decreased compared to non-DN patients [73]. The involvement of ferroptosis has also been confirmed in streptozotocin-induced DN animal models [72]. There were significant changes in ferroptosis-associated markers, including decreased expression levels GPX4 and increased expression of ACSL4 as well as lipid peroxidation products, and iron content in DN mice. Treatment with ferroptosis inducer, erastin or RSL3, could induce renal tubular cell death and increase the levels of iron and ACSL4 to sensitize those cells to ferroptosis [72]. Treatment with ACSL4 inhibitor, rosiglitazone, reduced lipid peroxidation product MDA and iron content in kidneys of DN mice, resulting in the improvement in survival rate and kidney function in those mice [72]. The role of ferroptosis in the development of DN is also supported by the finding that treatment with high glucose led to ferroptosis-specific mitochondrial changes in the HK-2 cells, whereas treatment with Fer-1 significantly rescued these changes and alleviated the renal pathological injuries in diabetic DN mice through Nrf2 pathway [74]. Treatment with Fer-1 also significantly ameliorated kidney hypertrophy and albuminuria to reduce the intracranial accumulation of lipid peroxidation in the diabetic mice by the HIF-1α/HO-1 pathway [75]. In addition, it has also been reported that HO-1 is expressed specifically in glomeruli in DN, and the induction of HO-1 prevents podocyte apoptosis [76,77]. Treatment with the antioxidant tempol increased the HO-1 activity in DN mice, resulting in the inhibition of oxidative stress and the restoration of redox balance [78]. Thus, induction of HO-1 may be beneficial for DN.

HMGB1 is a DNA-binding nonhistone protein and implicated in DNA replication, transcription, and DNA damage repair [79]. In addition to its nuclear functions, extracellular HMGB1 is a damage-associated molecular that triggers the inflammation and immunity response during ferroptosis induced by RSL3 and elastin in vitro [80]. In DN patients, HMGB1 are significantly elevated. Knockdown of HMGB1 suppressed high glucose-induced activation of TLR4/NF-κB axis and promoted Nrf2 expression as well as its downstream targets, including HO^−1^, NAD(P)H dehydrogenase (quinone 1) (NQO-1), glutamate–cysteine ligase catalytic subunit (GCLC), and glutamate–cysteine ligase modifier subunit (GCLM) in mesangial cells in vitro [81], which suggested that targeting HMGB1 for ferroptosis might be a novel therapeutic strategy in diabetic kidney disease.

Renal fibrosis is considered the common pathway of chronic progressive kidney disease [82]. Emerging evidence shows that ferroptosis is an integral process in the pathogenesis of renal fibrosis. TGF-β1 has been considered as a key mediator of renal fibrosis [83]. Treatment with TGF-β1 decreased the SLC7A11 and GPX4 expression in renal tubular cells and co-treatment with Fer-1 abrogates TGF-β1-induced cell death in vitro [73]. In doxorubicin induced-renal fibrosis animals, the non-heme iron levels, and the mRNA of renal prostaglandin–endoperoxide synthase 2 (PTGS2), a putative marker of ferroptosis, were increased in kidneys, which suggested a possible role of ferroptosis in renal fibrosis [84]. In unilateral ureter obstruction (UUO) or IRI indued fibrosis mouse kidneys, the expression of GPX4 was downregulated and the expression of 4-hydroxynonenal (4-HNE), the lipid peroxidation production, was upregulated [85,86]. In addition, the expression of HO-1 was peaked in mouse kidneys at 12 h after obstruction but decreased in kidneys one week after UUO [87]. Knockout of HO-1 (^−/−^) increased renal EMT and fibrosis as well as macrophage infiltration and the expression of renal tubular TGF-β 1 after 7 days of UUO compared to those in wild-type UUO mice [88]. Moreover, the induction of HO-1 inhibited the expression of TGF-β1 and proinflammatory molecules, and also reversed renal tubule-interstitial fibrosis [89]. Targeting ferroptosis with inhibitors, including Fer-1 and deferoxamine (DFO), could largely mitigate kidney injury, interstitial fibrosis, and inflammatory cell accumulation in mice after UUO or IRI injury [85,86]. In addition, treatment with tocilizumab mimotopes alleviates kidney injury and fibrosis by ferroptosis inhibition in a UUO model [90]. Iron chelation is considered as a possible treatment for renal fibrotic lesions [91]. In a CKD rat model, treatment with deferasirox (DFX), could mitigate renal fibrosis by inhibiting TGF-β1/Smad3, inflammation, and oxidative stress pathways [92]. Moreover, an iron-restricted diet exerts a renal protective effect by inhibiting oxidative stress and aldosterone receptor signaling in animal models of CKD [93,94].

### 3.3. The Roles of Ferroptosis in ADPKD

ADPKD is the most common inherited renal disease, which is caused by mutations of PKD1 (encoding polycystin-1) or PKD2 (encoding polycystin-2) and is characterized by multiple cysts in the kidneys which enlarge over time and lead to end-stage renal disease failure [95]. Cyst formation and progression in ADPKD are also involved in oxidative stress and inflammation and cell death [96,97,98]. It has been reported that the expression of antioxidant enzymes (such as GPX and SOD) and their activity were decreased in two different PKD animal models [99]. In addition, lipid peroxidation is increased in human polycystic kidneys and cyst growth is augmented by the lipid-peroxidizing compound tBHP in mouse embryonic kidney organ cultures [100]. Recently, we showed that inhibition of ferroptosis with Fer-1 delays cyst growth in rapidly and slowly progressive ADPKD mouse models, whereas induction of ferroptosis with its inducer, erastin, promotes cyst growth in those mouse models [15]. We provide evidence to support that low levels of cell death in kidneys from Pkd1 mutant mouse models are mainly ferroptotic, not apoptotic. The expression of the system Xc-, which are critical for the import of cystine (SLC7A11 and SLC3A2), the iron exporter (ferroportin), and GPX4 was decreased, and the expression of iron importers (TFR1, DMT1) and HO-1 was increased in Pkd1 mutant renal epithelial cells and tissues, resulting in high iron levels, low GSH and GPX4 activity, increased lipid peroxidation, and proneness to ferroptotic cell death. In addition, we showed that 4HNE, a lipid peroxidation product, is increased in Pkd1 null cells which is responsible to promote the proliferation of Pkd1 mutant cells via activation of Akt, S6, Stat3, and Rb. Collectively, our recent study suggests that ferroptosis is one of the important mechanisms to promote cyst progression in ADPKD and targeting ferroptosis may be a novel therapeutic strategy got ADPKD treatment [15].

### 3.4. The Roles of Ferroptosis in Renal Cell Carcinoma (RCC)

Kidney cancer, also known as renal cancer, is defined as a disease that starts in the kidneys. It is one of the top 10 most common cancers in the United States with more than 76,000 new cases diagnosed each year. A retrospective cohort study found that 26% of kidney cancer patients had CKD before tumor nephrectomy [101]. The cause of developing kidney cancer is still not clear. However, there are several factors that can increase the risk of kidney cancer, such as older age, smoking, obesity, hypertension, and being on kidney dialysis. RCC, also known as renal cell adenocarcinoma, is the most common type of kidney cancer in adults. A study demonstrated the risk of RCC in ESRD patients is increased up to 100 times [102]. RCC denotes cancer originated from the renal epithelium and accounts for >90% of cancers in the kidney and can be distinguished by histopathological features and gene mutations [103]. Localized RCC can be successfully manage with surgery, whereas metastatic RCC is refractory to conventional chemotherapy in which most patients show resistance to chemotherapy and radiotherapy. Activation of regulated cell death is considered an ideal therapeutic strategy for cancer and may help drug resistance. Research on the effect of erastin in 60 tumor cell lines of eight tissues found that RCC cells are more susceptible than others to erastin induced cell death [104]. Clear cell renal cell carcinoma (ccRCC) is the most common type of RCC. Both hereditary and familial ccRCC are strongly connected with von Hippel Lindau (VHL) gene mutations, which consecutively lead to the stabilization of hypoxia-inducible transcription factor (HIF) [105]. Miess et al. demonstrated that the silence of genes coding for glutathione peroxidases, GPX3 and GPX4, is lethal to ccRCC cells [106]. They also found that the cell viability of ccRCC is dependent on the synthesis of GSH, which prevents the accumulation of lipid peroxides. In addition, the re-expressed VHL gene had a resistance effect on ferroptosis in VHL-deficient RCC cells [106]. In addition, the susceptibility of ferroptosis can be affected by cell density and this effect is mediated by TAZ through the regulation of EMP1-NOX4 in ccRCC [107]. Furthermore, a 2–82% mutation rate among 36 ferroptosis associated genes (FRGs), including TP53, NFE2L2, FANCD2, DPP4, ALOX5, PTGS2, ALOX15B, ACSL4, CARS, and HMGCR, was detected in ccRCC in an analysis based on the GSCA database [108]. A new survival model was built based on five risk-related FRGs (CARS, NCOA4, FANCD2, HMGCR, and SLC7A11), which indicated that high expression of FANCD2, CARS, and SLC7A11 and low expression of HMGCR and NCOA4 are associated with high-risk ccRCC patients. These studies suggest that FRGs are the potential prognostic biomarkers and ferroptosis modulation may have therapeutic potentials in ccRCC.

Hereditary leiomyomatosis and renal cell cancer (HLRCC) are the autosomal dominant disorder caused by germline mutations in fumarate hydratase (FH), which is characterized by multiple cutaneous piloleiomyomas, uterine leiomyomas, and papillary type 2 renal cancer, which is resistant to current radiotherapy, chemotherapy, and immunotherapy [109]. It has been suggested that accumulated fumarate drives constitutive Nrf2 activation, which promotes the transcription of FTL and FTH1 genes in HLRCC cells in vitro [110]. On one side, excessive fumarate inhibits IRPs’ ability via the repression of FTL and FTH1 mRNA translation, which results in high intracellular ferritin levels [111]. High intracellular ferritin further sequesters free iron and finally results in a drop in the labile iron pool. On the other side, the FH accumulation sensitizes HLRCC cells to ferroptosis through C93 modification which represses GPX4 activity [112]. FH is also shown to indirectly inhibit AMPK, resulting in indirect inhibition of DMT1 expression [111]. This prevents the efflux of iron from the endosome into the cytoplasm to further reduce the labile iron pool. Induction of ferroptosis in FH-inactivated tumors represents an opportunity for synthetic lethality in cancer. Thus, pharmacological suppression of those proteins represents a treatment strategy worth exploring.

## 4. Targeting Ferroptosis for Kidney Disease Therapy

With the depth of research on ferroptosis, accumulating evidence indicates that ferroptotic cell death can inhibit tumor growth and improve the efficacy of chemotherapeutic drugs [113]. However, ferroptosis plays different roles in different kinds of kidney diseases. At present, preclinical studies have shown that ferroptosis can be successfully modulated in different kidney disease animal models and a variety of ferroptosis inducers and inhibitors have been administrated in those models (Figure 2). We summarize ferroptosis targeting agents with the therapeutic potential in kidney diseases in Table 2.

In AKI, ferroptosis may be through the recruitment of inflammation and other forms of regulated necrosis, leading to the amplification of kidney injury which suggests that inhibition of ferroptosis has therapeutic potential for the treatment of AKI diseases. In an I/R induced AKI animal model, treatment with ferroptosis inhibitor, 16–86 (a new third-generation ferrostatin), could protect acute tubular necrosis and I/R injury, also suggesting that ferroptosis is independent of the necroptosis-inhibiting complex in renal tubules, specifically Fas-associated protein with death domain and caspase-8 [114]. Treatment with liproxstatin-1, another ferroptosis inhibitor, could also ameliorate I/R-induced intestinal injury to prolong life in mice due to GPX4 deletion [30]. ACSL4 is a pivotal indicator and regulator of ferroptosis, which functions as a critical determinant of ferroptosis sensitivity by modulating the cellular (phospho) lipid composition. ACSL4 expression is upregulated under ischemic conditions and contributes to reperfusion-induced ferroptotic injury. Inhibition of ischemia-induced ACSL4 with rosiglitazone and siRNA decreased ferroptosis and lipid peroxidation, ameliorated cell damage, and intestinal barrier dysfunction caused by intestinal I/R in vivo and in vitro [115]. In addition, treatment with Fer-1 alleviated kidney injury and improved renal function in folic acid- and cisplatin-induced AKI mice [43,116].

In CKD, iron deposition initiates Fenton-mediated oxidative damage and further contributes to renal injury, suggesting that using iron chelators to deplete the labile iron pool for blocking lipid peroxidation is a potential therapeutic approach for the treatment of CKD. In CKD rats, DFX treatment can mitigate renal fibrosis by the inhibition of TGF-β1/Smad3, inflammation, and oxidative stress pathways [92]. Furthermore, the inhibitory effect of iron chelators on kidney fibrosis is also confirmed in UUO mice by alleviating iron metabolism and oxidative stress [85]. In addition to iron chelators, treatment with ferroptosis inhibitor Fer-1 also showed therapeutic potential for the treatment of CKD. In diabetic mice, Fer-1 treatment significantly ameliorated kidney hypertrophy and albuminuria and reduced the intrarenal accumulation of lipid peroxidation via the HIF-1α/HO-1 signaling pathway [114].

In ADPKD, although the roles of regulated cell death to cyst growth are controversial, several ferroptosis-targeted drugs are suggested to have therapeutic potential. It has been reported that exposure to scavengers of reactive oxygen species, such as glutathione, coenzyme Q10, or idebenone, blocks the growth of MDCK cysts by reducing the activation of TMEM16A (anoctamin 1) [111], in which TMEM16A has been demonstrated to be essential for regulating cyst growth [117]. We recently reported that treatment with ferroptosis inhibitor, Fer-1, delayed cyst growth in both early-stage and long-lasting ADPKD mouse models [15]. It has also been reported that Ferritin is markedly elevated in cystic kidneys of PKD mice [118]. Treatment with ciclopirox olamine (CPX-O), an iron chelator, inhibited ferritin accumulation in ADPKD kidneys and induced ferritinophagy in an iron-independent manner, resulting in the reduction in cyst growth in PKD mice [118]. These studies support that targeting ferroptosis may be a novel therapeutic strategy for ADPKD treatment.

In RCC, β-oxidation inhibition and fatty acid metabolism reduction make renal cancer cells highly dependent on the GSH/GPX pathway to prevent lipid peroxidation and cell ferroptotic death [119]. Hence, induction of ferroptosis has become a promising treatment for RCC. Deprivation of cystine induced rapid programmed necrosis in VHL-deficient cell lines and primary ccRCC cells in vitro, but not that in VHL-restored counterparts [120]. Deprivation of glutamine and cystine by addition of BSO (L-buthionine (S,R)-sulfoximine), which can inhibit GSH synthesis, sensitized ccRCC cell growth in a MYC-dependent RCC mouse model [106]. Sorafenib has been approved by the food and drug administration (Food and Drug Administration, FDA) for the treatment of multi-carcinoma, including RCC [121]. Moreover, apart from the typical ferroptosis inducer, compounds from traditional Chinese medicine, such as artesunate and lycorine, have also been found to inhibit the proliferation of RCC cells by the induction of ferroptosis in vitro [122,123]. Together, these studies suggest that targeting ferroptosis could be a promising strategy for the treatment of RCC.

At present, the effect of most of the ferroptosis inhibitors or inducers has been tested in in vitro experiments and rodent animal models with no measurable side effects. The efficacy of these drugs should be further evaluated in clinical settings for the treatment of kidney diseases, and a more comprehensive evaluation of the side effects of these drugs should be performed in the future.

## 5. Conclusions and Perspectives

Ferroptosis is a recently identified regulated cell death mediated by iron-dependent accumulation of lipid peroxidation. Ferroptosis is distinct from apoptosis, necrosis, and autophagy, based on the criteria of morphology (such as shrunken mitochondria), biochemical (such as inactivation of cellular GSH-dependent antioxidant defenses or loss of activity of GPX4), and genetic (such as numerous genes to regulate system Xc-, GSH, Gpx4, lipid peroxidation, and iron metabolism). We discussed the difference between ferroptosis and other regulated cell deaths, the signaling pathways involved in the regulation of ferroptosis, the role of ferroptosis in kidneys diseases, and the action of ferroptosis inhibitors in kidney diseases. We also discussed the epigenetic mechanisms in the regulation of ferroptosis, which may be a novel research direction and should be developed in the future. In addition, there are still some concerns that need to be addressed, including the following: (1) Although ferroptosis is morphologically, biologically, and genetically distinct from other types of cell death, such as apoptosis, autophagy, and necrosis, growing evidence indicates that ferroptosis is closely intertwined with all other forms of cell death to build a network. It is necessary to understand the molecular mechanisms of the crosstalk between ferroptosis and other forms of cell death and whether they are synergistic or antagonistic in kidney diseases, which may be exciting for the treatment of kidney diseases through targeting different forms of cell death. (2) Although diverse genes, such as ACSL4, GPX4, etc., have been found to regulate ferroptosis, none of those genes can be used as sensitive markers to monitor ferroptosis. It is necessary to identify specific markers for labeling ferroptosis in kidney diseases. (3) Because the in vitro cell culture conditions are quite different from in vivo conditions in animal models of kidney diseases, the interpretation of cell-based data and animal-based data related to the link between ferroptosis and kidney diseases, care must be taken. Currently, the link between ferroptosis and kidney diseases has been established only through cellular and animal experiments, which need to be further clarified by clinical trials. In summary, the activation and inhibition of ferroptotic-associated signaling pathways play an important role in kidney diseases. Further mechanistic and clinical understanding of the pathophysiological functions of ferroptosis will propose a promising targeted strategy for kidney diseases.

**Table 2 metabolites-12-00058-t002:** Pharmacological interference of ferroptosis in experimental kidney diseases.

Drugs	Effects	Experimental Model	Route and Dosage of Drug Administration	Local/ Systemic	Mechanisms of Different Drugs	References
**Ferrostain-1**	Prevented ROS formation	Animal model: Folic Acid-induced nephropathy; Cisplatin-induced nephropathy; UUO; Diabetic db/db mice; *Pkd1^RC/RC^* and *Pkd1^flox/flox^: Pkhd1-Cre* mice	IP: *Pkd1^RC/RC^* and *Pkd1^flox/flox^: Pkhd1-Cre* mice: 4 mg/kg/da; Cisplatin-induced nephropathy; 5 mg/kg/day Diabetic db/db mice; 1 mg/kg/day; UUO: 5 mg/kg	systemic	Inhibition of the upregulation of IL-33; Inhibition HIF-1α/HO-1 pathway; Normalize the iron metabolism and inhibiting cell proliferation through Akt, S6, Stat3 and Rb signal pathways	[15,43,75,116]
**SRS 16-86**	Ferroptosis inhibition	Animal model: IRI model	IP: 2 mg/kg	systemic	Inhibition of mitochondrial permeability transition, postischemic and toxic renal necrosis.	[114]
**DFX/DFO**	Chelate iron	Animal model: UUO; 5/6 nephrectomy	DFO: IP: 100 mg/kg/day DFX: Oral: low dose 15 mg/kg/day, moderate dose 30 mg/kg/day high dose 60 mg/kg/day	systemic	TGF-β1/Smad3, inflammation, and oxidative stress pathways	[85,92]
**Liproxstain-1**	Ferroptosis inhibition	Animal model: IRI model	IP: 10 mg/kg	systemic	Induces GPX4 expression, reduces COX2 expression and inhibition of the kidney inflammation activation	[115]
**N-acetyl-l-** **cysteine**	Reduce ROS	Animal model: Cisplatin-induced nephropathy	IP: 50 mg/kg	systemic	Inhibition of the kidney inflammation activation and the complement system	[124]
**Rosiglitazone**	Decrease ROS	Animal model: STZ-induced diabetic kidneys	Intragastric: 5 or 20 mg/kg/day Oral: 3 mg/kg/day	systemic	Inhibition of NF-ĸB activation and MCP-1 expression.	[125,126]
**CPX-O**	Chelate iron	Animal model: *Pkd1*^RC/RC^/*Pkd2*^+/−^ mice	IP: 10 mg/kg/day	systemic	Induces ferritin degradation via ferritinophagy	[118]
**L-buthionine** **(S,R)-sulfoximine**	Deprivation of glutamine and cystine	Animal model: MYC-dependent RCC mouse model	Oral: 20 mM in drinking water/day	systemic	GSH synthesis inhibition	[106]
**Artesunate**	Induction of ferroptosis	Cell model: Sunitinib-resistant RCC cells	20 uM in cultured media	local	Cell cycle arrest and modulation of cell cycle regulating proteins	[122]

CPX-O: Ciclopirox olamine; DFX: Desferriox; DFO: Deferroxamine; ROS: reactive oxygen species; HIF-1α: hypoxia-inducible factor 1-alpha; HO-1: heme oxygenase-1; UUO: unilateral ureteral obstruction; IRI: renal ischemia–reperfusion injury; CKD: chronic kidney disease; GPX4: glutathione peroxidase 4; COX2: cyclooxygenase-2; MCP-1: monocyte chemoattractant protein-1; RCC: renal cell carcinoma; GSH: glutathione.

## Figures and Tables

**Figure 1 metabolites-12-00058-f001:**
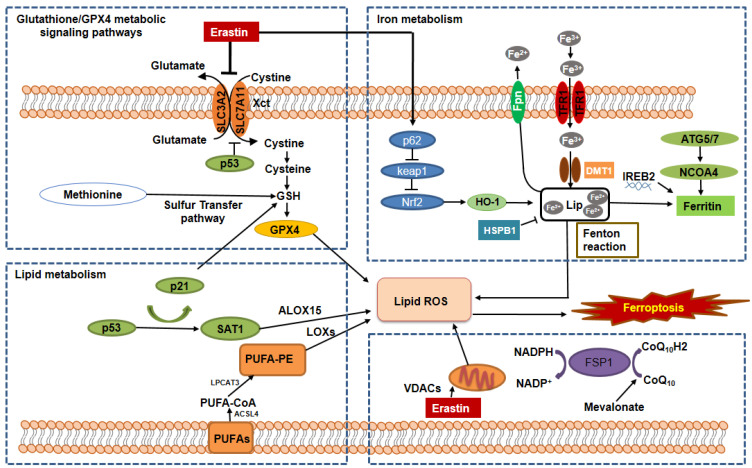
**The mechanisms and signaling pathways regulate the metabolisms of glutathione/GPX4, iron, and lipid in ferroptosis.** The hallmarks of ferroptosis include the loss of lipid peroxide repair capacity by the phospholipid hydroperoxidase GPX4, the availability of redox-active iron, and oxidation of PUFA-containing phospholipids, which are regulated by the following signaling pathways. 1. Glutathione/GPX4 signaling pathways. The synthesis of tripeptide GSH protects cells from ferroptotic death, in that GSH is essential: (i) for the maintenance of the level of GPX4 through the exchange of glutamate and cystine via cystine/glutamate antiporter system Xc-, and (ii) for the functional activity of GPX4 to reduce lipid hydroperoxides (L-OOH) to lipid alcohols (L-OH), resulting in the prevention of the iron (Fe^2+^)-dependent formation of toxic lipid ROS. The inhibition of system Xc- or inhibition of GSH synthesis should result in lipid peroxides accumulation and ferroptotic cell death through inactivating GPX4. In addition, the mevalonate pathway, sulfur transfer pathway, glutamine pathway, and p53 signaling axis are also involved in the regulation of GPX4 activity and function. 2. Iron metabolic signaling pathway. Iron metabolism includes iron uptake (transferrin receptor), iron export (ferroportin), iron storing (ferritin), and ferritinophagy mediated by lysosome and NCOA4. Abnormal iron metabolism could induce iron overload resulting in lipid ROS production via the Fenton reaction. In addition, ferritin can be regulated by the ATG5-ATG7 and NCOA4 pathways as well as and IREB2. Furthermore, iron metabolism can be regulated by p62-Keap1-NRF2 and HSPB1 signaling pathways. 3. Lipid metabolic signaling pathway. Ferroptosis can be caused by an abnormal accumulation of lipid peroxidation, which is the archetype free radical chain reaction formally resulting in the insertion of O_2_ into a C-H bond in the oxidizable free PUFAs. PUFAs can be transformed into PUFA-PEs by two enzymes, including ACSL4 for synthesizing PEs and LPCAT3 for lipid remodeling. PUFAs can also be oxidized to phospholipid hydroperoxides by LOXs. In addition, recent studies show that VDACs and the FSP1-CoQ10-NAD(P)H pathways exist as an independent parallel system that works cooperatively with GPX4/glutathione to regulate lipid peroxidation and ferroptosis. GSH: glutathione; TFR1: transferrin receptor 1; Fpn: ferroportin; NCOA4: nuclear receptor coactivator 4; DMT1: divalent metal transporter 1; ATG5: autophagy protein 5; ATG7: autophagy-related protein 7; IREB2: iron-responsive element-binding protein 2; HO-1: heme oxygenase 1; Nrf2: nuclear factor-erythroid factor 2-related factor 2; LIP: labile iron pool; PUFAs: polyunsaturated fatty acids; ACSL4: acyl-CoA synthetase long-chain family member 4; LPCAT3: lysophosphatidylcholine acyltransferase 3; PEs: phosphatidylethanolamines; LOXs: lipoxygenases; ALOX15: arachidonate lipoxygenase 15; VDACs: voltage-dependent anion channels.

**Figure 2 metabolites-12-00058-f002:**
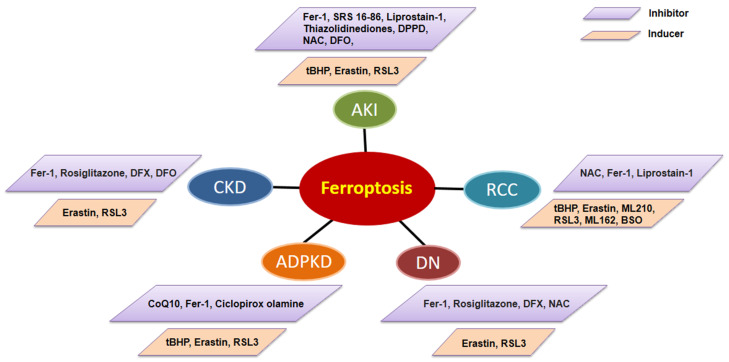
The ferroptosis inducers and inhibitors used/tested in animal models of kidney disorders, including AKI, DN, CKD, ADPKD, and RCC. AKI: acute kidney injury; DN: diabetic nephropathy; CKD: chronic kidney disease; ADPKD: autosomal dominant polycystic kidney disease; RCC: renal cell carcinoma. Fer-1: Ferrostain-1; DPPD: diphenyl-p-phenylenediamine; NAC: N-acetyl-l-cysteine; ferric ammonium citrate, DFO: deferoxamine mesylate; Rosiglitazone: the inhibitor of ACSL4; DFX: deferasirox; CoQ10: Coenzyme Q10; BSO: L-buthionine (S,R)-sulfoximine.

**Table 1 metabolites-12-00058-t001:** The morphological and biochemical features of the different types of cell death.

	Ferroptosis	Apoptosis	Autophagy	Necroptosis
**Morphological** **Features**	Cell membrane: lack of rupture and blebbing of the plasma membrane, rounding-up of the cell Cytoplasm: small mitochondria with increased mitochondrial membrane densities, reduction or vanishing of mitochondria crista, outer mitochondrial membrane rupture Nucleus: normal nucleus	Cell membrane: formation of apoptotic bodies and cytoskeletal disintegration Cytoplasm: cellular volume reduction but no significant changes in mitochondrial structure Nucleus: nuclear volume reduction; chromatin agglutination; nuclear fragmentation	Cell membrane: normal cell membrane Cytoplasm: formation of double-membraned autolysosomes, including macroautophagy, microautophagy and chaperone-mediated autophagy Nucleus: lack of chromatin condensation,	Cell membrane: plasma membrane breakdown Cytoplasm: generalized swelling of the cytoplasm and organelles Nucleus: moderate chromatin condensation
**Biochemical** **Features**	Iron accumulation and lipid peroxidation; Inhibition of system Xc- with decreased cystine uptake; Release of arachidonic acid mediators	Activation of caspases oligonucleosomal DNA fragmentation	Increased lysosomal activity (e.g., LC3-I to LC3-II conversion)	Drop in ATP levels; Activation of RIP1, RIP3, and MLKL; Release of DAMPs; PARP1 hyperactivation
**Core proteins**	SLC7A11, Nrf2, HO-1, GPX4, p53, TFR1, VDAC2/3, ACSL4, LOXs	p53, Bax, Bak, Bcl-2 family proteins	ATG5, ATG7, Beclin 1 and other ATG family proteins	RIP1, RIP3, MLKL

System Xc-: cysteine/glutamate transporter receptor; LC3: microtubule-associated protein 1 light chain 3; ATP: adenosine triphosphate; RIP1: receptor-interacting protein 1; RIP3: receptor-interacting protein 3; MLKL: mixed lineage kinase domain-like protein; DAMPs: damage-associated molecular patterns; PARP1: poly (ADP-ribose) polymerase 1; SLC7A11: solute carrier family 7 member 11; Nrf2: nuclear factor erythroid 2-related factor 2; HO-1: heme oxygenase-1; GPX4: glutathione peroxidase 4; TFR1: transferrin receptor 1; VDAC2/3: voltage-dependent anion channels 2/3, ACSL4: acyl-CoA synthetase long-chain family member 4; LOXs: arachidonate lipoxygenase; Bax: BCL2-associated X protein; Bak: BCL-2 homologous antagonist/killer; ATG5: autophagy protein 5; ATG7: autophagy-related 7.

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
