# Peer review of "Abnormal Iron and Lipid Metabolism Mediated Ferroptosis in Kidney Diseases and Its Therapeutic Potential"

_metabolites, 2022, doi:10.3390/metabo12010058_

Round 1

Reviewer 1 Report

The manuscript by Zhang and Li reviewed the ferroptosis-related molecules and their associations with renal diseases. The manuscript is well organized and summarized. However, there are some concerns needed to be addressed.  

Major concerns:

  1. In line 438, “These findings suggest an essential role of GPX4 in kidney injury” intends to interpret GPX4 as a key molecule to cause AKI. Please rephrase it.
  2. In the necroptosis part: RIP1 (receptor-interacting protein 1) and RIPK1 (receptor-interacting protein kinase -1) are the same molecules, as well, RIP3 (receptor-interacting protein 3) and RIPK3 (receptor-interacting protein kinase-3) are the same molecules. Please unify them.
  3. This review article aims to present the association of ferroptosis and kidney injury, which expects to provide a new insight view in the treatment of renal diseases. The article is well summarized in ferroptosis regulation. However, less than half paragraphs discuss the association of kidney injury and ferroptosis in this manuscript. In the 3rd part, the regulation of ferroptosis by epigenetic modification is seemed to be redundant in this manuscript. It is suggested to re-edit it.  
  4. The review article is entitled “Abnormal amino acid, iron, and lipid metabolism mediated ferroptosis in kidney diseases and its therapeutic potential”. However, no information about amino acid homeostasis linking to kidney injury is discussed in this manuscript. It is suggested to re-edit the title.

Minor concerns:

  1. There are some typing errors:

-       Line 123, 321: transferrin receptor 1(TFR1)

-       Line 202: RIP3: receptor-interacting protein 3

-       Line 298, BAP1 is not BAPI.

-       Figure 3 and Table 2, liproxstatin-1

Please check the spelling.

  1. The full name of Bax is BCL2-associated X protein and the full name of Bak is BCL-2 homologous antagonist/killer, BCL2 antagonist/killer 1, or BCL2-like 7 protein. Please correct them.
  2. In Figure 1, p21 and CDKN1A are the same molecules. Please delete one.
  3. The abbreviations of ATG5 and ATG7 are different in this manuscript. NRF2 or Nrf2. Please unify them.

Author Response

Major concerns:

  1. In line 438, “These findings suggest an essential role of GPX4 in kidney injury” intends to interpret GPX4 as a key molecule to cause AKI. Please rephrase it.

Response: Thank you for your suggestion. We rephrased this sentence as suggested in the revised manuscript, which was marked in red in line 298.

  1. In the necroptosis part: RIP1 (receptor-interacting protein 1) and RIPK1 (receptor-interacting protein kinase -1) are the same molecules, as well, RIP3 (receptor-interacting protein 3) and RIPK3 (receptor-interacting protein kinase-3) are the same molecules. Please unify them.

Response: Thank you for your comments and suggestion. We unified them in the revised manuscript, which were marked in red in Table 1.

  1. This review article aims to present the association of ferroptosis and kidney injury, which expects to provide a new insight view in the treatment of renal diseases. The article is well summarized in ferroptosis regulation. However, less than half paragraphs discuss the association of kidney injury and ferroptosis in this manuscript. In the 3rd part, the regulation of ferroptosis by epigenetic modification is seemed to be redundant in this manuscript. It is suggested to re-edit it.  

Response: Thank you for your comments and suggestion. We deleted the section of the regulation of ferroptosis by epigenetic modification. We added more information about the association of kidney injury and ferroptosis in the revised manuscript, which were marked in red.

  1. The review article is entitled “Abnormal amino acid, iron, and lipid metabolism mediated ferroptosis in kidney diseases and its therapeutic potential”. However, no information about amino acid homeostasis linking to kidney injury is discussed in this manuscript. It is suggested to re-edit the title.

Response: Thank you for your suggestion. We re-edited our title as “Abnormal iron and lipid metabolism mediated ferroptosis in kidney diseases and its therapeutic potential”

Minor concerns:

  1. There are some typing errors:

-       Line 123, 321: transferrin receptor 1(TFR1)

-       Line 202: RIP3: receptor-interacting protein 3

-       Line 298, BAP1 is not BAPI.

-       Figure 3 and Table 2, liproxstatin-1

Response: Thank you for your suggestion. We corrected all the typing errors in the revised manuscript.

Please check the spelling.

  1. The full name of Bax is BCL2-associated X protein and the full name of Bak is BCL-2 homologous antagonist/killer, BCL2 antagonist/killer 1, or BCL2-like 7 protein. Please correct them.

Response: Thank you for your suggestion. We corrected those as suggested in the revised manuscript and marked those modifications in red.

  1. In Figure 1, p21 and CDKN1A are the same molecules. Please delete one.

Response: Thank you for your suggestion. We modified as suggested.

  1. The abbreviations of ATG5 and ATG7 are different in this manuscript. NRF2 or Nrf2. Please unify them.

Response: Thank you for your suggestion. We unified those in the revised manuscript.

Reviewer 1

The manuscript by Zhang and Li reviewed the ferroptosis-related molecules and their associations with renal diseases. The manuscript is well organized and summarized. However, there are some concerns needed to be addressed.  

Major concerns:

  1. In line 438, “These findings suggest an essential role of GPX4 in kidney injury” intends to interpret GPX4 as a key molecule to cause AKI. Please rephrase it.

Response: Thank you for your suggestion. We rephrased this sentence as suggested in the revised manuscript, which was marked in red in line 298.

  1. In the necroptosis part: RIP1 (receptor-interacting protein 1) and RIPK1 (receptor-interacting protein kinase -1) are the same molecules, as well, RIP3 (receptor-interacting protein 3) and RIPK3 (receptor-interacting protein kinase-3) are the same molecules. Please unify them.

Response: Thank you for your comments and suggestion. We unified them in the revised manuscript, which were marked in red in Table 1.

  1. This review article aims to present the association of ferroptosis and kidney injury, which expects to provide a new insight view in the treatment of renal diseases. The article is well summarized in ferroptosis regulation. However, less than half paragraphs discuss the association of kidney injury and ferroptosis in this manuscript. In the 3rd part, the regulation of ferroptosis by epigenetic modification is seemed to be redundant in this manuscript. It is suggested to re-edit it.  

Response: Thank you for your comments and suggestion. We deleted the section of the regulation of ferroptosis by epigenetic modification. We added more information about the association of kidney injury and ferroptosis in the revised manuscript, which were marked in red.

  1. The review article is entitled “Abnormal amino acid, iron, and lipid metabolism mediated ferroptosis in kidney diseases and its therapeutic potential”. However, no information about amino acid homeostasis linking to kidney injury is discussed in this manuscript. It is suggested to re-edit the title.

Response: Thank you for your suggestion. We re-edited our title as “Abnormal iron and lipid metabolism mediated ferroptosis in kidney diseases and its therapeutic potential”

Minor concerns:

  1. There are some typing errors:

-       Line 123, 321: transferrin receptor 1(TFR1)

-       Line 202: RIP3: receptor-interacting protein 3

-       Line 298, BAP1 is not BAPI.

-       Figure 3 and Table 2, liproxstatin-1

Response: Thank you for your suggestion. We corrected all the typing errors in the revised manuscript.

Please check the spelling.

  1. The full name of Bax is BCL2-associated X protein and the full name of Bak is BCL-2 homologous antagonist/killer, BCL2 antagonist/killer 1, or BCL2-like 7 protein. Please correct them.

Response: Thank you for your suggestion. We corrected those as suggested in the revised manuscript and marked those modifications in red.

  1. In Figure 1, p21 and CDKN1A are the same molecules. Please delete one.

Response: Thank you for your suggestion. We modified as suggested.

  1. The abbreviations of ATG5 and ATG7 are different in this manuscript. NRF2 or Nrf2. Please unify them.

Response: Thank you for your suggestion. We unified those in the revised manuscript.

Reviewer 2 Report

This manuscript entitled "Abnormal amino acid, iron, and lipid metabolism mediated ferroptosis in kidney diseases and its therapeutic potential" is very well written by Zhang and Li. They provide very comprehensive molecular mechanisms of ferroptosis and its correlation with kidney disease. I especially like the way they discuss ferroptosis in different type of kidney diseases. The authors also pointed out critical inducers and inhibitors of ferroptosis for each kidney disease model, which will be very useful for readers. I recommend the manuscript to be accepted for publication.

I have a few minor comments that the authors may consider revising their manuscript.  

  1. line 87, "exit" and "entry" of glutamate (Glu) and cystine (Cys) respectively...
  2. line 99, and studied "enzymes"
  3. line 143, "the insertion of O2 into a C-H bond", Would it be possible to rephrase this sentence? It is not the best way to describe the chemical reaction.
  4. line 222, "Fourth, is" probably missing something in this sentence.
  5. It is not clear why epigenetic modification as a separate section instead of part ferroptosis mechanism (in section 2). Does epigenetic modification particularly important for kidney disease? 

Author Response

  1. line 87, "exit" and "entry" of glutamate (Glu) and cystine (Cys) respectively...

Response: Thank you for your suggestion. We corrected those in the revised manuscript and marked those modifications in red.

  1. line 99, and studied "enzymes"

Response: Thank you for your suggestion. We corrected as suggested in the revised manuscript.

  1. line 143, "the insertion of O2 into a C-H bond", Would it be possible to rephrase this sentence? It is not the best way to describe the chemical reaction.

Response: Thank you for your suggestion. We modified this sentence as suggested.

  1. line 222, "Fourth, is" probably missing something in this sentence.

Response: Thank you for your suggestion. We modified this sentence in the revised manuscript.

  1. It is not clear why epigenetic modification as a separate section instead of part ferroptosis mechanism (in section 2). Does epigenetic modification particularly important for kidney disease? 

Response: Thank you for your suggestion. We deleted the section of the regulation of ferroptosis by epigenetic modification in the revised manuscript.

Reviewer 3 Report

The work includes an extensive and updated summary of recent studies on ferroptosis in humans, organized by the different pathways that participate in the ferroptosis. The final part is focused on the role of ferroptosis in kidney disease, by considering separately the acute, chronic, polycystic kidney diseases and renal carcinoma, together with the potential of anti-ferroptosis drugs. I found that this part is rather confused, superficial, and not well organized. In general not very convincing of the importance of ferroptosis in kidney diseases. A reorganization of the paragraphs is needed with more details on the models and treatments. 

- the papers that deal with ferroptosis in kidney diseases are quoted, but the models are not well explained, and it is not clear if they are cell or animal models. It would be better to keep the two well separated.

- To support the hypothesis of the importance of some factors, some unrelated models are quoted, increasing the confusion. Fore example at line 443-447 Alzheimer's and hepatocellular carcinoma is mentioned to stress the importance of HO-1 in AKI. And at L. 504-506 human leukemia is mentioned.

- In some examples, iron deregulation is considered as ferroptosis, as in paragraphs 460-467. In general, a clear definition of ferroptosis should be applied, and the experimental details described. 

- I do not find particularly useful Table 2 and accompanying paragraph 5, since no detail on the use of various drugs is shown. How was the drug administered, local or systemic, on cells or animal, dosage, effect, and side effect?

- There are some English errors that would easily be corrected by a writing assistant. 

Author Response

  1. The papers that deal with ferroptosis in kidney diseases are quoted, but the models are not well explained, and it is not clear if they are cell or animal models. It would be better to keep the two well separated.

Response: Thank you for your comment and suggestion. We modified as suggested to better indicate the in vitro and in vivo animal models in the revised manuscript.

  1. To support the hypothesis of the importance of some factors, some unrelated models are quoted, increasing the confusion. For example at line 443-447 Alzheimer's and hepatocellular carcinoma is mentioned to stress the importance of HO-1 in AKI. And at L. 504-506 human leukemia is mentioned.

Response: Thank you for your comment and suggestion. We revised the manuscript by deleting those models unrelated to kidney diseases.

  1. In some examples, iron deregulation is considered as ferroptosis, as in paragraphs 460-467. In general, a clear definition of ferroptosis should be applied, and the experimental details described.

Response: Thank you for your comment and suggestion. We re-defined ferroptosis and added more information in the revised manuscript, which were marked in red.

  1. I do not find particularly useful Table 2 and accompanying paragraph 5, since no detail on the use of various drugs is shown. How was the drug administered, local or systemic, on cells or animal, dosage, effect, and side effect?

Response: Thank you for your comment and suggestion. We modified Table 2 by adding the details on the application of various drugs in different animal models of kidney diseases, which were marked in red.

  1. There are some English errors that would easily be corrected by a writing assistant.

Response: Thank you for your comment and suggestion. We corrected all those errors in the revised manuscript.

Round 2

Reviewer 3 Report

The manuscript improved. 

However, in table 2 DFX was given IP and DFO oral.  Is the opposite, since DFX is the oral chelator.